

# Sensitivity of warm clouds to large particles in measured marine aerosol size distributions – a theoretical study

Tom Dror[1,*], J. Michel Flores[1,*], Orit Altaratz[1], Guy Dagan[2], Zev Levin[3], Assaf Vardi[4], and Ilan Koren[1]

[1]Department of Earth and Planetary Sciences, Weizmann Institute of Science, Rehovot, Israel.
[2]Atmospheric, Oceanic and Planetary Physics, Department of Physics, University of Oxford, Oxford, UK.
[3]School of Earth Sciences, Department of Geophysics, Tel Aviv University, Ramat Aviv, Israel.
[4]Department of Plant and Environmental Sciences, Weizmann Institute of Science, Rehovot, Israel.
[*]Contributed equally to this work.

**Correspondence:** Ilan Koren (Ilan.koren@weizmann.ac.il)

**Abstract.** Aerosol size distribution has major effects on warm cloud processes. Here, we use newly acquired marine aerosol size distributions (MSD), measured *in-situ* over the open ocean during the Tara Pacific expedition (2016-2018), to examine how the total aerosol concentration ($N_{tot}$) and the shape of the MSD change warm cloud's properties. For this, we used a toy-model with detailed bin-microphysics. The changes in the MSDs affected the clouds' total mass and surface precipitation. In

general, the clouds showed higher sensitivity to changes in $N_{tot}$ than to changes in the MSD's shape, except for the case where the MSD contained giant and ultragiant cloud condensation nuclei (GCCN, UGCCN). For increased $N_{tot}$, most of the MSDs drove an expected non-monotonic trend of mass and precipitation. However, the addition of GCCN and UGCCN drastically changed this trend, such that surface rain saturated and the mass monotonically increased with $N_{tot}$. GCCN and UGCCN changed the interplay between the microphysical processes by triggering early initiation of collision-coalescence. The early

fall-out of drizzle in those cases enhanced the evaporation below the cloud base. Testing the sensitivity of rain yield to GCCN and UGCCN revealed an enhancement of surface rain upon the addition of larger particles to the MSD, up to a certain particle size, when the addition of larger particles resulted in rain suppression. This finding suggests a physical lower bound can be defined for the size ranges of GCCN and UGCCN.

**1   Introduction**

Clouds play a key role in the Earth's climate system. By scattering and absorbing solar and terrestrial radiation, clouds influence the radiative balance. Aerosols influence cloud processes by serving as cloud condensation nuclei (CCN) on which cloud droplets can form (Köhler, 1936). The size of CCN determines the droplets' initial size distribution and hence impacts cloud processes and properties, such as size (Rosenfeld et al., 2008; Altaratz et al., 2014; Koren et al., 2014), lifetime (Albrecht,

1989), optical properties (Twomey and Squires, 1959; Twomey, 1977; Mülmenstädt and Feingold, 2018), and rain amounts





and patterns (Yin et al., 2000b; Rosenfeld et al., 2006; Xue et al., 2008; Yuan, 2011; Altaratz et al., 2014; Koren et al., 2014; Seigel, 2014).

The study of giant CCN (GCCN) and ultragiant CCN (UGCCN) and their effects on warm clouds and precipitation have been the subject of various works (Beard and Ochs III, 1993; Feingold et al., 1999; Khain et al., 2000; Yin et al., 2000b; Dagan

et al., 2015a). Their size definition is loose, as the lower threshold of GCCN has been defined within a wide range of mean particle diameter ($D_p$) of $2 - 10\,\mu m$ (Feingold et al., 1999; Yin et al., 2000a), while particles with $D_p > 20\,\mu m$ are usually defined as UGCCN (Feingold et al., 1999; Posselt et al., 2008). Although their observed concentration is low ($< 0.1\,cm^{-3}$; Exton et al., 1986; Flores et al., 2020) in comparison to a typical marine CCN concentration ($50 - 250\,cm^{-3}$), they have been shown to affect cloud properties, and might even transform non-precipitating clouds to a precipitating state (Feingold et al.,

30  1999).

GCCN and UGCCN stem from a variety of sources, but are considered to be mainly sea-salt (Schulz et al., 2004) and mineral dust (Levin et al., 1996; Tegen et al., 2002). Despite their large size, these particles can be transported thousands of kilometers from their origin. Ultragiant mineral dust particles ($D_p > 75\,\mu m$) have been observed as far as $10,000\,km$ from their origin (Betzer et al., 1988). Other studies have shown even bigger dust particles ($D_p > 200\,\mu m$) carried from Asia to

the remote Pacific Ocean, and from the Sahara to Europe (Middleton et al., 2001). Recently, gigantic Saharan dust particles ($D_p \sim 450\,\mu m$) were observed above the Atlantic Ocean $\sim 3,500\,km$ west of the African coast (van der Does et al., 2018).

Aerosols' ability to act as CCN is largely controlled by their size (Dusek et al., 2006), thus, even though mineral dust is less soluble than sea-salt (Petters and Kreidenweis, 2007; Kumar et al., 2009), large mineral dust particles are still considered to act as effective GCCN (Johnson, 1982; Levin et al., 1996; Nenes et al., 2014).

The effect of GCCN and UGCCN on warm clouds' processes is highly important but not fully understood. Early work demonstrated that a few activated UGCCN, and even GCCN (from $\sim 10^{-3}\,cm^{-3}$) can drive early initiation of precipitation, by producing a tail of large drops in the droplet distribution (Johnson, 1982). More recent studies have shown that the effect of GCCN and UGCCN on warm clouds and precipitation is more complex and greatly depends on aerosol concentration. For low aerosol concentration, the addition of GCCN was shown to have little or no effect on precipitation (Teller and Levin, 2006;

Zhang et al., 2006; Cheng et al., 2009; Dagan et al., 2015a), due to the early initiation of collision-coalescence and lower supersaturation values (Zhang et al., 2006). In contrast, their effect under polluted conditions is still under debate. It is accepted that the addition of small CCN (for constant liquid water content) leads to the formation of a greater number of smaller droplets, and results in delayed collision-coalescence and a less efficient collection process (Gunn and Phillips, 1957; Squires, 1958; Warner, 1968; Albrecht, 1989). However, addition of GCCN and UGCCN, on one side, has been shown to counteract this

delay and act to precede and enhance the collection process, leading to earlier initiation of precipitation (Johnson, 1982; Teller and Levin, 2006; Feingold et al., 1999; Yin et al., 2000b; Rosenfeld et al., 2002; Zhang et al., 2006; Cheng et al., 2009; Dagan et al., 2015a). This was demonstrated for warm convective clouds (Cheng et al., 2009; Dagan et al., 2015a) and stratiform clouds (Feingold et al., 1999; Zhang et al., 2006). On the other side, Khain et al. (2000) reported that the role of GCCN and UGCCN, though it can be important, is unlikely to be the dominant mechanism of raindrop formation in warm clouds. On a

global scale, by using the ECHAM5 General Circulation Model, Posselt et al. (2008) found that adding GCCN induces faster





precipitation in warm clouds, and shorter residence times and less accumulation of water in the atmosphere (i.e., accelerating the hydrological cycle).

Here we present a theoretical study, combining new *in-situ* measurements of marine aerosol size distributions (MSD), taken during the *Tara* Pacific expedition (Flores et al., 2020), and a "toy-model" with a detailed description of cloud microphysical
processes, to examine the link between MSD and cloud processes and properties (like cloud mass and amount of precipitation), on a single-cloud scale. By using a simplified model, we gain the ability to distill the MSD effect on the interplay between the cloud microphysical processes. This study can be viewed as a basis for a future investigation of this effect on a cloud field scale.

## 2   Methods

### 2.1   MSD Measurements

MSDs were measured aboard the schooner *Tara* over the Atlantic Ocean, Caribbean Sea, and Pacific Ocean during the *Tara* Pacific Expedition (2016-2018). The Pacific Expedition primary focus was coral reef research (Planes et al., 2019) with the supporting measurements of discrete surface ocean measurements (Gorsky et al., 2019), and the innovative addition of marine aerosol measurements (Flores et al., 2020). Using a scanning mobility particle sizer (SMPS) in parallel with an optical particle
counter (OPC), particles between $0.03 - 32\,\mu m$ (dry diameter) were measured at $\sim 15\,m$ above sea level (ASL) in the Atlantic Ocean and at $\sim 27\,m$ ASL in the Caribbean Sea and western Pacific Ocean (Fig. 1). A Nafion dryer was installed before the SMPS-OPC, which reduced the sampled air relative humidity to below $\sim 35\%$, below the efflorescence point for NaCl (Gupta et al., 2015), thus we considered the particle diameter as dry. The OPC size distributions were corrected and merged with the SMPS size distributions following the method described by Hand and Kreidenweis (2002). For a more detailed description of
the aerosol measurements see Flores et al. (2020). Six MSDs were chosen for this study to initiate the cloud simulations (Fig. 1b): two from the Atlantic Ocean, one from the Caribbean Sea, and three from the Pacific Ocean.

The MSDs represent a variety of marine environments with different scenarios: *Atlantic-1*, anthropogenically influenced, with a single mode located between the Aitken and Accumulation modes, highly pronounced coarse and giant modes, and total aerosol concentration ($N_{tot}$) of $2629\,cm^{-3}$; *Atlantic-2*, with comparable Aitken and Accumulation modes, pronounced coarse mode,
and no giant mode ($N_{tot} = 416\,cm^{-3}$); *Caribbean-3*, with comparable Aitken and Accumulation modes, a less pronounced coarse mode, and no giant mode ($N_{tot} = 677\,cm^{-3}$); *Pacific-4*, anthropogenically influenced single mode, a less pronounced coarse mode, and no giant mode ($N_{tot} = 4193\,cm^{-3}$); *Pacific-5*, clean marine with a more pronounced Accumulation mode, a diminished coarse mode, and no giant mode ($N_{tot} = 168\,cm^{-3}$); and *Pacific-6*, super clean marine with a more pronounced Aitken mode, a diminished coarse mode, and no giant mode ($N_{tot} = 89\,cm^{-3}$).

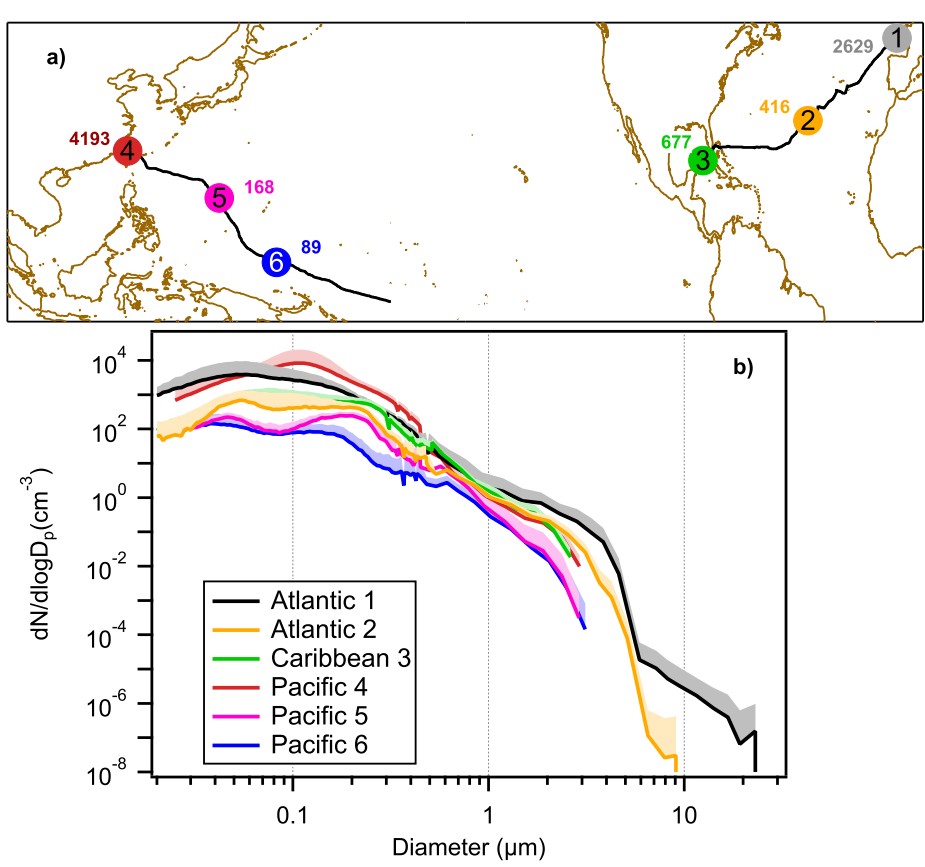

**Figure 1.** (a) *Tara's* route across the Atlantic Ocean, Caribbean Sea and the Pacific Ocean. Circles indicate the locations of the MSDs used in this study, with total number concentrations written next to each circle. (b) All MSDs; shaded areas represent the upper standard deviation. Each colored curve in (b) is associated with a specific location and total concentration marked in the same color in (a). Each MSD is an average of at least eight hours of measurements.





### 2.1.1 Model Description and Setup

The Tel Aviv University axi-symmetric (1.5 D; vertical and radial directions) non-hydrostatic cloud model (TAU–CM) with a detailed cloud microphysics scheme was used (Tzivion et al., 1994; Reisin et al., 1996). The TAU-CM includes warm microphysical processes such as nucleation of CCN, condensation and evaporation, collision-coalescence, breakup (McTaggart-Cowan and List, 1975; Low and List, 1982), and sedimentation (cold processes were excluded here). The microphysical processes are formulated and solved using a multi-moment bin method (Tzivion et al., 1987). CCN of a certain size are activated if the critical supersaturation is reached according to the Köhler equation (Pruppacher and Klett, 1980), taking into account both the curvature and chemical (i.e., solute) effects. All the MSDs were considered to be composed of sea-salt aerosols. To test the sensitivity of the results to different chemical composition, we ran extra simulations changing the aerosol's composition to ammonium sulfate, and found no substantial differences.

The model was run at $50\,m$ resolution in the vertical and horizontal directions, and a temporal resolution of $1\,s$. The model was initialized using three idealized atmospheric profiles. We chose to use the idealized profiles since the MSDs were sampled throughout different places (see Fig. 1), and our focus is on the MSD's effect. The profiles represent a relatively moist tropical environment (Garstang and Betts, 1974; Dagan et al., 2015b), but differ in the inversion layer height and the humidity in the cloudy layer. The deepest profile included a well-mixed sub-cloud layer between $0 - 1000\,m$, and a conditionally unstable cloudy layer between $1000$ and $4000\,m$ ($3000, 2000\,m$ for the other profiles) with relative humidity (RH) of 95% (90, 80%). The cloudy layers were bounded by an overlying inversion layer with a temperature gradient of $2^o$C over $50\,m$, and RH of 30%. Here we focus on the deepest profile (highest inversion height and RH), and present some of the results from the other two profiles in the supplementary information (SI). This choice was made because a larger aerosol concentration optimum is expected for larger clouds (Dagan et al., 2015b). This allowed us to examine the full effect of the different MSDs on cloud microphysical processes. Each of the six MSDs was normalized to the five other MSD concentrations, to preserve the original shape (see Fig. S1; total of 36 MSDs and 108 simulations for three initialization profiles).

### 3 Results and Discussion

First, we explored the link between the MSD and the cloud's bulk properties (total mass and rain yield) as a function of the total aerosol concentration ($N_{tot}$).

Figure 2 presents the total accumulated rain yield at the surface (Fig. 2a) and the maximum cloud mass for each simulation (Fig. 2b) as a function of the $N_{tot}$ used in that simulation. Each curve presents the results of six different simulations conducted using the same MSD shape but with different concentration (each MSD was normalized to the concentration of the other MSDs while maintaining its shape). As can be seen in Fig. 2 (and in Fig. S2 for the two other atmospheric profiles), the *Atlantic-1* clouds have a distinct curve compared to the rest of the MSDs for all profiles. We will first describe the curves of the other five MSD clouds, and later focus on the exceptional *Atlantic-1* curve and its driving mechanisms.

The general shape of the five curves is similar, and exhibits a non-monotonic trend (Fig. 2a,b, deepest profile): an increase in total rain yield and the cloud's maximum mass as a function of aerosol loading, up to a maximum optimal aerosol concentration

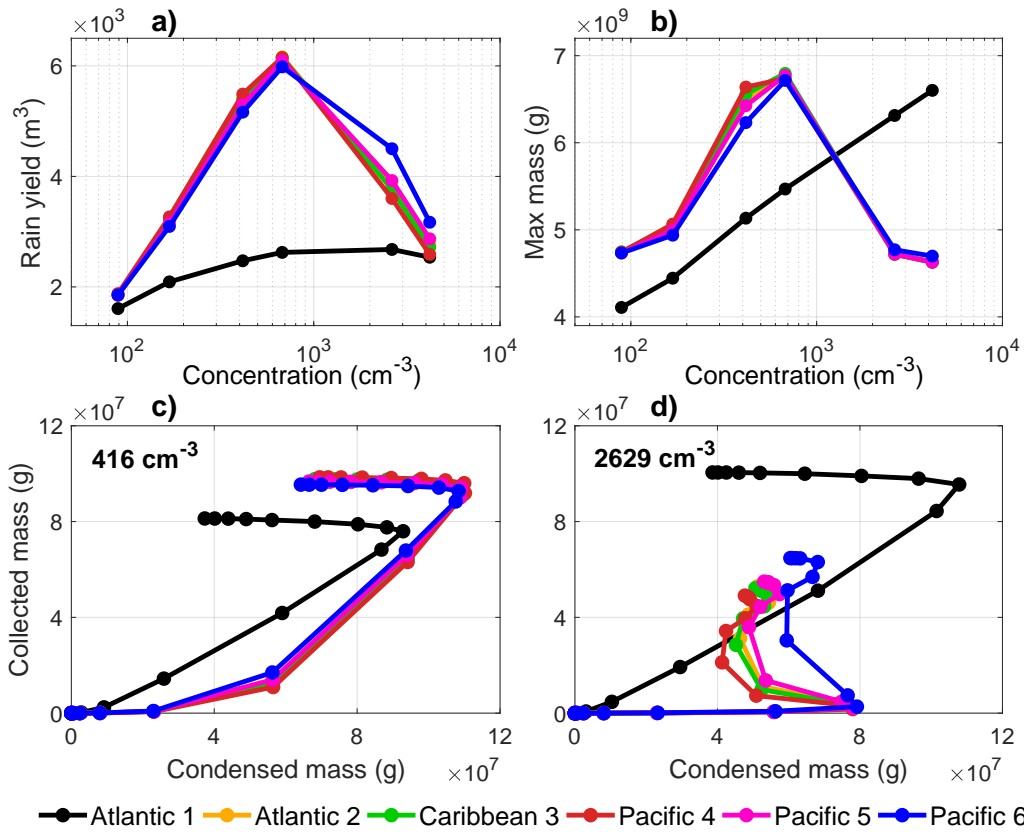

**Figure 2.** (a) Surface rain yield and (b) cloud's maximum total mass as a function of aerosol concentration used in the simulation. Each curve represents six simulations, done with a specific shape of the MSD normalized to different aerosol concentrations. The lower panels (c, d) present the time evolution (5-min time steps are marked by filled circles) of accumulated condensed mass versus accumulated collected mass. The panels represent an aerosol concentration of 416 and $2629\,cm^{-3}$ (c and d, respectively).

($N_{op}$), followed by a decrease. All five curves have similar $N_{op}$ (around $N_{tot} = 677\,cm^{-3}$) for both surface rain yield and maximum cloud mass. This non-monotonic trend can be explained by the interaction of competing processes (Dagan et al., 2015b). The ascending branch (moving from extremely clean to slightly polluted conditions) can be explained by the droplets' increased surface area, which enhances condensation efficiency (Pinsky et al., 2013; Seiki and Nakajima, 2014) and delays the initiation of collision-coalescence (see Fig. S3a,b). The delayed initiation of collision-coalescence drives longer condensational growth (hence, latent heat release also increases), and the droplets reach higher in the atmosphere (larger droplet mobility; Koren et al. (2015)). This chain of processes drives deeper clouds with more liquid mass (i.e., cloud invigoration). On the other hand, the descending branch ($N_{tot} > N_{op}$) is caused by enhanced periphery processes, entrainment and evaporation, which take over and result in cloud suppression (see evaporation in Fig. S3c and Dagan et al. (2015b)). The value of $N_{op}$ depends on the atmospheric profile (Dagan et al., 2015b), such that it decreases as the profile becomes shallower (i.e., lower inversion





base and RH; Fig. S2). For the cases of shallower profiles, where the clouds are more subjected to entrainment effects, the ascending branch of the curves is less pronounced (Fig. S2a,b) or non-existent (Fig. S2c,d). We therefore focus on the deepest

atmospheric profile, which better demonstrates the full effect of the competition and interactions between the microphysical processes in the clouds.

The curve formed by the *Atlantic-1* MSD clouds (black line in Fig. 2) is dramatically different from the other five curves. It shows not only significantly lower values for most of the runs (except for the cloud mass for $N_{tot} > \sim 1000\,cm^{-3}$), but also different trends in both rain yield and cloud mass. For $N_{tot} < 1000\,cm^{-3}$, the trends of both surface rain and the cloud's

maximum mass show an increase with increasing aerosol loading, similar to the other five MSD curves. However, for higher values of aerosol concentration, the rain yield saturates, and the cloud's maximum mass continues to increase with no $N_{op}$ (for higher aerosol concentration values see Fig. S4). The flattening of the rain yield curve is attributed to the presence of GCCN (Dagan et al., 2015a). The *Atlantic-1* MSD has three distinct modes: one influenced by pollution, one of coarse particles, and one of GCCN. Next, we examined how the coarse and giant modes, which are by far more pronounced in the *Atlantic-1* MSD

than in the other five MSDs, account for the unique behavior of these clouds.

To explore the *Atlantic-1* MSD monotonic increase in maximum cloud mass with $N_{tot}$, we examined the time evolution of the cloud's microphysical processes. Figure 2c,d shows the evolution of the six cloud trajectories on the phase space spanned by the accumulating condensed mass (representing droplet nucleation and condensational growth) and the accumulating collected mass (representing the collision-coalescence processes), for a medium aerosol concentration level ($N_{tot}$ normalized to

$416\,cm^{-3}$, Fig. 2c) and a more polluted one ($N_{tot}$ normalized to $2629\,cm^{-3}$, Fig. 2d). Note that the collected mass represents an internal redistribution of the liquid water mass with no change in the total mass. For the cleaner cases, where the aerosol loading is a bit lower than $N_{op}$, in the first stage, all but the *Atlantic-1* clouds, accumulate mass by nucleation and condensation without any contribution from the collection process. At a later stage in the cloud's lifetime, the trajectories turn diagonally up, showing that the collection process has begun. Finally, the clouds stop growing by condensation, reaching their maximum

mass, and begin to evaporate (trajectories turn to the left). In the *Atlantic-1* MSD case, the collection process kicks in earlier, within 10 minutes of the cloud's lifetime, due to the presence of GCCN in the MSD which initially form bigger droplets. The bigger droplets resulted in a lower total droplets' surface area for the *Atlantic-1* MSD case compared with the other MSDs (Fig. S5), which was then further reduced by the early initiation of the collection process. Moreover, these bigger droplets rapidly grow into drizzle-sized drops and sediment out of the cloud (see below), accounting for the smaller maximum total condensed

mass in the *Atlantic-1* case compared to the other five cases.

Under more polluted conditions, the trajectory of the *Atlantic-1* MSD cloud (black curve in Fig. 2d) on this phase space is similar (in shape) to the one in the cleaner case (black curve in Fig. 2c), but this cloud accumulates more mass, due to the larger droplet surface area (Fig. S5b,c). The more polluted the clouds are, the greater the difference in the total droplet surface area between the *Atlantic-1* MSD and the rest of the MSDs. The *Atlantic-1* cloud also condenses more mass compared

to the other MSDs (reaching $\sim 11 \times 10^7\,g$ compared to $\sim 8 \times 10^7\,g$). This is explained by the fact that the *Atlantic-1* cloud nucleates much more on the GCCN, and hence accumulates more mass. The accumulation of liquid water by nucleation and condensation occurs in parallel to the collection process that starts much earlier in this case (Fig. 2d). In the other five MSD





clouds, collision-coalescence begins toward the end of the condensational growth stage, or after the evaporation process has begun (e.g., *Pacific-4*, the trajectories turn back to the left before acquiring a vertical component). The timing of the initiation

of collision-coalescence further explains the decreasing branch of the rain yield trend for all MSDs aside from *Atlantic-1* (Fig. 2a), as it starts too late in the cloud's lifetime (after the cloud has already begun to lose mass).

This, however, does not explain the overall smaller surface rain yield of the *Atlantic-1* MSD clouds for all aerosol concentrations, or the saturation trend for high $N_{tot}$ (Fig. 2a). To further inspect the lower surface rain yield, we examined the temporal evolution of evaporation below cloud base, and the surface rain rate for the different MSD clouds. Figure 3 shows the evap-

oration below cloud base (left column), and the surface rain rate (right column) as a function of time for four different $N_{tot}$ ($89, 416, 2629, 4193\,cm^{-3}$ – from upper to lower panels).

Two main features can be seen for the *Atlantic-1* case, regardless of $N_{tot}$: (i) both evaporation below cloud base and surface rain start earlier than in the rest of the cases. This is due to the early onset of collision-coalescence (Fig. 2c,d), which converts the already big particles into drizzle-sized drops; (ii) evaporation below cloud base is always larger and, at the same time, the

surface rain rate is lower. However, the magnitude of both evaporation below cloud base and surface rain rate does depend on $N_{tot}$, ranging from $0.17 - 2.16 \times 10^{7}\,(g\,s^{-1})$ and $0.42 - 4.76 \times 10^{7}\,(g\,s^{-1})$, respectively.

The smaller values of the surface rain yield for the *Atlantic-1* MSD, and the non-monotonic trend for the rest of the MSDs (Fig. 2a), are also evident in the temporal evolution of surface rain rate (right column of Fig. 3). Part of this is explained above, by the interplay between different internal cloud processes, but it does not elucidate the complete mechanism. Figure 3 shows

that evaporation below cloud base plays a crucial role in determining the low values of the surface rain in the *Atlantic-1* case. As $N_{tot}$ increases, more GCCN are present (Fig. S1) and preferentially activated, growing rapidly into drizzle-sized drops, which immediately begin to precipitate. This reduces their time spent in the cloud, and they are thus large enough to fall, but still too small to reach the surface before they fully evaporate. The *Atlantic-1* raindrops are considerably smaller than those produced by the other clouds (Fig. S6), and their evaporation is therefore greater. Greater evaporation below cloud base can

lead to a larger descent of cold air to the surface and eventually, to cold pool formation, which affects cloud field organization (Warner et al., 1979; Zuidema et al., 2012; Seifert and Heus, 2013; Dagan et al., 2018).

Finally, to ensure that this reported effect is indeed a direct result of the presence of GCCN, we investigated the impact of the different parts of the *Atlantic-1* MSD on cloud processes. We performed additional sensitivity simulations using the *Atlantic-1* MSD in which the largest aerosol size bins were gradually excluded from the distribution. This resulted in a very minor change

in $N_{tot}$ ($< 0.001\%$), due to the small number concentration of the excluded large particles.

Figure 4a presents the total surface rain yield for a specific simulation as a function of the aerosol threshold diameter used in that simulation (above which the particle concentration was set to zero). For example, $D_9$ represents a simulation in which the *Atlantic-1* MSD was truncated at an aerosol size of $9\,\mu m$ (i.e., all size bins with diameters larger than this threshold were set to zero). The behavior of the surface rain yield as a function of threshold diameter (Fig. 4a) revealed that the amount of pre-

cipitation reaching the surface is highly dependent on the existence of GCCN, and more specifically on their sizes. The curve shows a non-monotonic trend that starts with a plateau, where the addition of larger particles (increase in threshold diameter) does not affect the surface precipitation ($D_{1.03} - D_{5.81}$, green shading). This stable behavior is followed by a range of sizes,



**Figure 3.** Time evolution of the evaporated mass below cloud base and surface rain mass per unit time (left and right columns, respectively). Each row represents a specific aerosol concentration: $89\,cm^{-3}$ (a,b), $416\,cm^{-3}$ (c,d), $2629\,cm^{-3}$ (e,f), and $4193\,cm^{-3}$ (g,h), as shown in the upper right corner of each row. The different curves in each panel represent an MSD shape normalized to the specific aerosol concentration. Note that there is an order of magnitude difference between the exponent in the right and left columns.

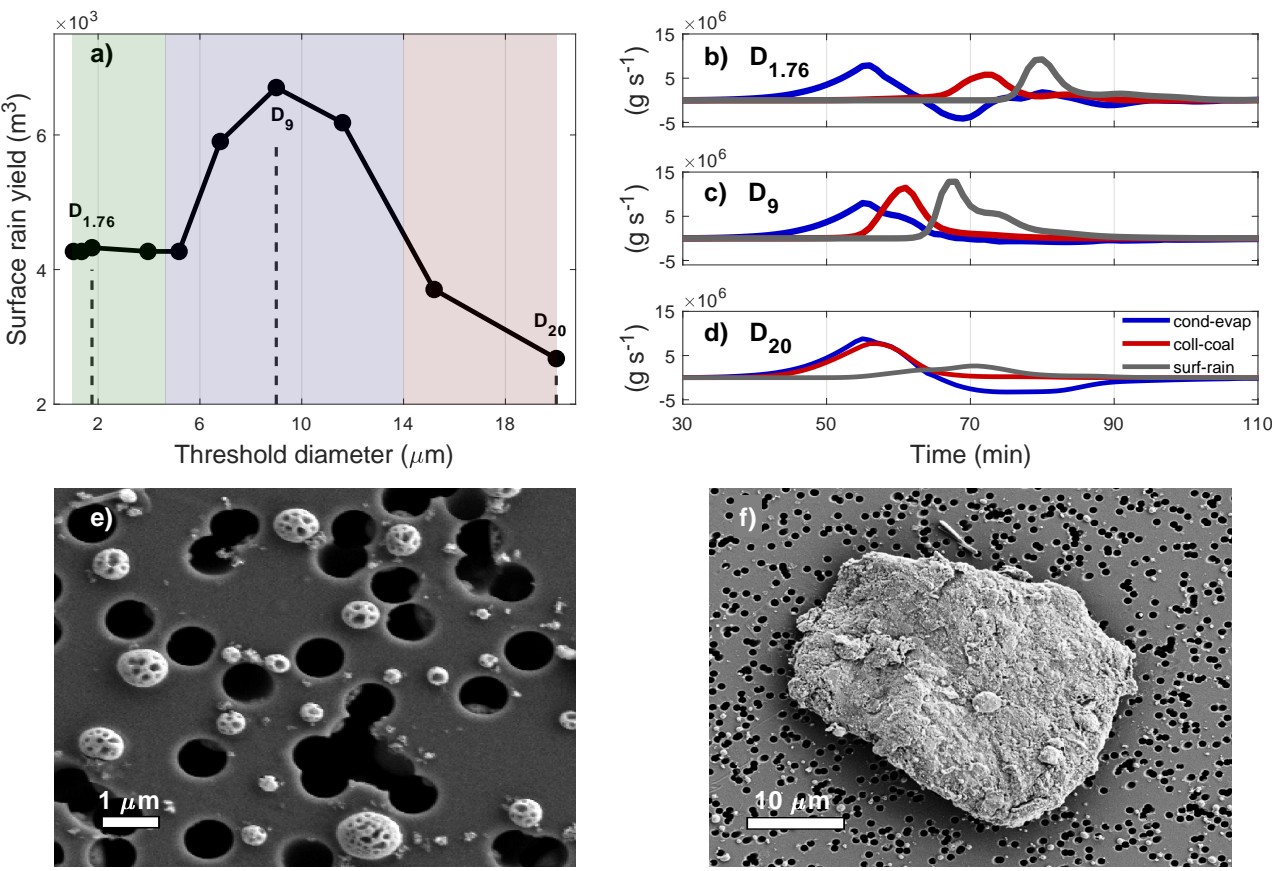

**Figure 4.** (a) Surface rain yield ($m^3$) as a function of a threshold diameter (no particles bigger than that diameter) for the *Atlantic-1* MSD. Each circle in (a) represents a subtraction of specific bins from the MSD. The shaded areas indicate the different trends of the curve. (b–d) Condensed–evaporated mass (blue), collected mass (red), and surface rain rate per unit time (gray), as a function of time (upper right panels: b–d) for selected threshold diameters matching three simulations from (a). The specific threshold diameters are marked in the upper left corner of each panel ($D_{1.76}$, $D_9$, and $D_{20}$ $\mu m$: b-d, respectively), and (d) represents the full *Atlantic-1* MSD. Scanning electron microscope images of pollen-like particles (e), and mineral dust (f) collected during the same period as the *Atlantic-1* MSD measurement.





where the addition of particles results in an enhanced amount of surface rain ($D_{5.81} - \sim D_{14}$, blue shading). The maximum surface rain yield ($6.7 \times 10^3 \, m^3$) is obtained at an optimum threshold diameter ($D_{op}$) of $\sim 9 \, \mu m$. As particles greater than $\sim 14 \, \mu m$ in diameter were included in the MSD, the rain yields decreased below the mean values of the plateau ($D_{14} - \sim D_{20}$, red shading). The changing trends of the curve suggest that the threshold used to define GCCN and UGCCN can be taken from a more physical source, rather than a loose definition. We propose that the threshold diameter for which the surface rain yield is enhanced be defined as the lower bound for GCCN, and the threshold diameter for which surface rain begins to be suppressed as the lower bound for UGCCN. For this study, using the *Atlantic-1* MSD and the specific atmospheric conditions (described in section 2.1.1), the lower bound of GCCN is $D_p \cong 5 \mu m$, and for UGCCN $D_p \cong 14 \, \mu m$.

Figure 4b,d presents the evolution (timing and magnitude) of the condensation–evaporation processes (nucleation and diffusional growth), collision-coalescence, and surface rain rate for three selected threshold diameters ($D_{1.76}$, $D_9$ and $D_{20}$). It sheds light on the different trends shown in Figure 4a. The larger the particles in the MSD, the faster the critical size for the initiation of the collision-coalescence process is reached, and the sooner it occurs. The initiation of collision-coalescence shortens from $\sim 65$ minutes of simulation for the $D_{1.76}$ case to $\sim 45$ minutes of simulation for the $D_{20}$ case, where it starts almost immediately after condensation begins (Fig. 4b and d, respectively).

For optimal rain production, collision-coalescence has to be correctly timed with the condensational growth of the cloud (Dagan et al., 2015a). For the $D_{1.76}$ case, collision-coalescence starts only after condensational growth has ceased, and peaks when evaporation is the dominant process ($\sim 62$ and $\sim 72$ minutes into the simulation, respectively). Whereas, for $D_9$ it starts earlier ($\sim 55$ minutes into the simulation) while condensation peaks, and for the $D_{20}$ case (i.e., the full *Atlantic-1* MSD), the peaks of the collision-coalescence and condensation processes occur at nearly the same time ($\sim 56$ minutes into the simulation). The optimum threshold diameter dictates the correct timing for the microphysical processes, such that maximum liquid water mass is converted to surface rain. For the *Atlantic-1* MSD (under the deepest atmospheric profile), the maximum surface rain is obtained for $D_{op} \sim 9 \, \mu m$ (Fig. 4c).

## 4 Summary

In this study, we used six MSDs measured *in-situ* in the Atlantic Ocean, Caribbean Sea and Pacific Ocean to study the effect of aerosol concentration and size on warm clouds' properties. The MSDs differed in shape and ranged in total aerosol concentration from very clean ($89 \, cm^{-3}$, *Pacific-6*) to polluted ($4193 \, cm^{-3}$, *Pacific-4*) conditions. By equating the $N_{tot}$ of the different MSDs (i.e., normalizing them to match the six specific $N_{tot}$) we altered their total aerosol concentration, while keeping the amount of small versus big aerosols constant. This affected the initial droplet size distributions in terms of the total number of droplets, and the droplets' sizes.

Using an axisymmetric cloud model with detailed bin–microphysics, we examined the sensitivity of key properties of warm clouds (cloud maximum mass and surface rain) to the measured MSDs on a single cloud scale, under a range of environmental conditions (going from shallow to deeper conditions, using three atmospheric profiles). We focused on the deepest profile, since it best captured the effect of competing microphysical cloud processes, and showed that surface rain yield and cloud's





maximum mass are affected in a non-monotonic way by changes in $N_{tot}$, and the shape of the MSDs for most of the cases. All MSD shapes, except for the *Atlantic-1*, shared a similar trend as a function of $N_{tot}$, starting with an increase in cloud mass and surface rain yield up to an $N_{op}$ of $\sim 700\, cm^{-3}$, followed by a decrease for higher aerosol loading. This consistent be-havior was altered by the increased concentration of giant particles in the *Atlantic-1* MSD. Namely, the maximum cloud mass

monotonically increased as a function of $N_{tot}$, while the surface rain yield also increased but then saturated at high aerosol concentrations (with no $N_{op}$). The surface rain yield also had lower values in all cases, dropping by a factor of up to 2.3. The former can be explained by efficient nucleation of the big aerosols, and the latter can be explained by the initiation time of collision-coalescence with respect to the optimal timing for accumulation of enough water by condensation, enabling more water to become available for rain production.

In addition, the immediate sedimentation post-nucleation produced small drizzle droplets that fall early, but evaporate below the cloud base before they reach the surface. Although the MSDs differed throughout the entire spectrum of aerosol sizes, this study shows that it is the existence of the giant mode that dramatically changes cloud properties, especially with respect to surface precipitation.

A deeper investigation of the effect of GCCN was preformed by gradually eliminating the largest particles from the *Atlantic-1*

MSD. We found that above a threshold diameter of $\sim 5\,\mu m$, collision-coalescence begins earlier, such that the surface rain is enhanced. This behavior is disrupted when the threshold diameter reaches $\sim 14\,\mu m$, with a further increase in threshold diameter resulting in lower surface rain yield. The rain suppression observed from this threshold diameter on is explained by the dramatically reduced droplet surface area, and the initiation of collision-coalescence at a much earlier stage. This results in fast formation of large drops and thus, an early fall-out of drizzle. These two values of threshold diameter are suggested to

define the lower bounds of GCCN and UGCCN. They depend on the specific conditions: atmospheric profile, $N_{tot}$ and the shape of the aerosol size distribution.

The *Atlantic-1* MSD was measured off the coast of France (see Fig. 1a). Scanning electron microscope images of the aerosols collected during the same time period highlight the differences between the particles. From a giant mode comprised of $\sim 1\,\mu m$ brochosomes-like particles (Fig. 4e), to an ultragiant mode comprised of $\sim 20\,\mu m$ mineral dust particles (Fig. 4f). Here, we

considered only changes in $N_{tot}$ and the MSD's shape. In addition, we also examined the sensitivity of the results to different chemical composition, which in the TAU–CM model affected the Köhler activation. Future work is needed to further explore how the chemical composition of the particles affects warm cloud's properties.

This study demonstrates the importance of the aerosol size distribution in terms of both total number concentration and the aerosol distribution shape, which can impact cloud properties. Currently, most aerosol measurements restrict the upper

limit of particle sizes to $D_p = 10\,\mu m$ (i.e., $PM_{10}$). Consequently, most of the cloud-resolving models, even those using bin-microphysics, do not allow for ultragiant or even giant particles. Many of these models use a "typical" 'wide-marine' or 'narrow-continental' size distribution that does not account for the natural variability in aerosol size distributions or reflect their complexity.



*Data availability.* Key parameters from this study are included in the SI, and will be uploaded to a public repository.

*Author contributions.* J.M.F and G.D conceived the presented idea. T.D. lead the simulations and G.D. supported. J.M.F. performed the measurements. All authors provided critical feedback and helped shape the research, analysis and manuscript. T.D. and J.M.F. took the lead in writing the manuscript, and contributed equally to this manuscript.

*Competing interests.* The authors declare no competing interests.

*Acknowledgements.* This research was partially supported by a research grant from Scott Jordan and Gina Valdez, the De-Botton center for Marine Science, the Yeda-Sela center for basic research, the Sustainability and Energy Research Initiative (SAERI).
We are keen to thank the following institutions for their financial and scientific support that made the unique Tara Pacific expedition possible: CNRS, PSL, CSM, EPHE, Genoscope/CEA, ANR, agnes b., the Veolia Foundation, Region Bretagne, Billerudkorsnas, Amerisource Bergen Company, Altran, Lorient Agglomeration, Prince Albert II de Monaco Foundation, L'Oreal, Biotherm, France Collectivites, FFEM, the Tara
Foundation teams, crew and board members. Tara Pacific would not exist without the continuous support of the participating institutes. The authors also particularly thank Serge Planes, Denis Allemand and the Tara Pacific consortium. This is publication number 14 of the Tara Pacific Consortium.



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
