# Peer review of "Sensitivity of warm clouds to large particles in measured marine aerosol size distributions – a theoretical study"

_Atmospheric Chemistry and Physics, 2020_

## Short Comment (SC1) · 10 Jun 2020

Very nice work, Fo a more complete understanding of aerosol types I would suggest you make mention of atmospheric microplastics including their sizes. The field is quite new however data is expanding and it could well play an important part in cloud formation. I suggest Allen et al. 2020 https://journals.plos.org/plosone/article?id=10.1371/journal.pone.0232746 and keep an eye out for Brahney et al (currently under embargo) as the numbers could be very useful to you. Best of luck.

[Figure]

2020.

---

## Referee Comment (RC1) · Anonymous Referee #1 · 27 Jun 2020

In this paper, an axisymmetric cloud model with detailed bin–microphysics was initialized with six marine aerosol size distributions (MSD), measured in-situ in the Atlantic Ocean, Caribbean Sea and Pacific Ocean to study the effect of aerosol concentration and size distribution on warm clouds' properties. It shows that the cloud mass and precipitation change non-monotonically with the total concentration and size distribution of cloud condensation nuclei (CCN), especially when a tail of giant or ultragiant CCN (GCCN or UGCCN) is also included in the aerosol size distribution. The most interesting finding is the upper boundaries of the GCCN. This has not been reported in previous studies, to the best of my knowledge. The study is well within the scope of ACP and is generally well presented, except for a few places need to be clarified or

corrected.

Specific comments:

1) In the abstract, the statements and explanations are mainly based on the simulation results using the deepest thermodynamic profile, a more generalized statement or results including the shallower clouds should also be included, for a more complete picture.

2) I suggest each of filled circles in Fig. 2 (c)(d) to be marked with a time. The current figures are a little bit confusing.

3) Line 183-184: "The Atlantic-1 raindrops are considerably smaller than those produced by the other clouds (Fig. S6), and their evaporation is therefore greater": One fact might also be important is that while raindrops formed earlier in case Atlantic-1, the cloud is still in its developing stage, or the vertical velocity is still positive below cloud base, and the relative humidity is relatively low, so the raindrops spend more time and therefore evaporate more before reaching the surface. In other cases, rain was promoted by stronger downdrafts, and the relative humidity should also be higher. Therefore, it may provide more evidence to explain the differences in surface rain amount and evaporation mass between case Atlantic-1 and other cases by analyze the below-cloud vertical velocity and relative humidity.

4) Figure 4(e) and 4(f) show a little bit strange here and do not add more support to the main body of the text, may be removed?

5) In this paper, only results from the simulation with the most unstable thermodynamic profile are analyzed in detail. For shallower clouds, the rain yield and the max. cloud mass show monotonic change with CCN concentration and no significant changes with GCCN. So the conclusion should be generalized to reflect how the results change for other thermodynamic situations.

Technical corrections:
1) Line 143, change "accumulating" to "accumulated";

2) Line 206: Figure 4b,d should be "Figures 4b-d.

---

## Author Comment (AC1) · 16 Jul 2020

Dear Steve Allen, thank you very much for having read our work and for your important comment. Traditionally, the sources of GCCN and UGCCN were considered mainly sea-salt and mineral dust. But you are right, it has now become clear that atmospheric microplastic is also an important source. After reading your work, and the recent study by Brahney et al., we are going to mention atmospheric microplastics and their sizes in to the paper. Many thanks.

---

## Referee Comment (RC2) · Anonymous Referee #3 · 14 Sep 2020

The influence of aerosol size distribution and chemical composition on precipitation formation and intensity is still a challenging question to answer, due primarily to the sophisticated microphysical processes dealing with particles with a wide range of sizes, and also to the interplays between dynamics and microphysics. In this study, the authors choose to focus on addressing aerosol-precipitation response in a warm cloud, using a detailed bin microphysical framework for both aerosols and cloud droplets while a somewhat simplified dynamical framework (an axisymmetric model). In addition, they have also assumed a uniform chemical composition for the included aerosol population (sea salt) to limit the aerosol activation in a one-dimensional (size) parametric space. In order to address the targeted issue more realistically, they have also adopted

measured aerosol size distributions collected from locations with different atmospheric backgrounds.

An interesting finding of this study is the significant difference in aerosol-precipitation responses between a case with the so-called Atlantic-1 aerosol profile with ultra large CCNs and cases with other measured aerosol profiles without evident fraction of such giant CCNs. With a careful design of their modeling simulations, the authors have been able to define the criterion size of large aerosol particles that can create significant impacts on precipitation. Overall speaking, the paper has been relatively well organized, the research findings are well presented, and conclusions are drawn with solid science evidence.

A clear missing information in the manuscript is the cloud droplets concentrations, especially the vertical profiles of number concentration of cloud droplets and raindrops. The authors have discussed the correlation between sub-cloud evaporation and rainfall at surface. With a knowledge of sub-cloud raindrop population including total number and size distribution this would be much easier to understand. In addition, Fig. 2(d) presents a rather interesting feature in high concentration simulations using all the distributions except Atlantic 1 where collision-coalescence overwhelmed the condensation growth in a relatively early stage. However, without information of vertical distributions of cloud mass, the reader would have problem to understand (1) why the collision-coalescence increases with time but in a rather slow pace comparing to the case of Atlantic-1, and (2) the depths of layer where cloud mass grew in various cases. Note that large droplets (i.e., raindrops) can still be moved upward by updraft and both condensation and collision-coalescence can proceed in either updraft or downdraft (as far as the parcel remains saturated). Therefore, knowledge of the vertical growth tracks of precipitating particles is critical to understand how the two major growing processes evolved.

It is understood that the authors wanted to focus on the aerosol and cloud microphysical connections. Nevertheless, the feedback of dynamics, even in a rather simplified

dynamical framework still plays a role in determining the growth of precipitating particles. The authors mentioned very briefly about cold downdraft and also analyzed sub-cloud evaporation. Perhaps a more in-depth analysis would provide a better understanding of the role of dynamical feedback in, e.g., leading to the results presented in Fig. 3 and 4.

Some minor comments.

Page 5, Figure 2(a) and (b): it would be helpful to provide the integration length of each simulation shown in these two figure panels in the figure caption.

Page 5, Ln 112: I understand the purpose of normalizing every distribution to match a given total concentration is for the convenience to identify the role of certain characteristics of size distribution such as shape in influencing the formation of precipitation. However, it is expected that the shift of the distributions to meet often much higher concentrations would increase the number of GCCN or even UGCCN. Could the authors provide such numbers even in the supplementary materials as a table or so? In addition, I don't remember this has been discussed in the manuscript, e.g., why the increase of GCCN still had no effect on the overall rain formation and growth for all cases including Atlantic-2 other than Atlantic-1.

Page 7, Ln 152: "bigger droplets resulted in a lower total droplets' surface area...", the sentence is somewhat ambiguous since such a result is not obvious, an explanation would be helpful here.

---

## Author Comment (AC2) · 11 Oct 2020

Dear Steve Allen, Thank you very much for having read our work and for your important comment. We have added a phrase to the summary of the paper regarding the issue of microplastic particles. The last paragraph of the paper now reads: "This study demonstrates the importance of the aerosol size distribution in terms of both total number concentration and the aerosol distribution shape, which can impact cloud properties. Currently, most aerosol measurements restrict the upper limit of particle sizes to Dp = 10 $\mu$m (i.e., PM10). Consequently, most of the cloud-resolving models, even those using bin-microphysics, do not allow for ultragiant or even giant particles. Many

of these models use a "typical" 'wide-marine' or 'narrow-continental' size distribution that does not account for the natural variability in aerosol size distributions or reflect their complexity. Additionally, with the mounting evidence of microplastic particles, with sizes between 4 – 188 $\mu$m, present in the atmosphere and in rain (Allen et al., 2020; Brahney et al., 2020) it is of greater importance to include and further study the impact of particles with Dp > 10 $\mu$m on clouds and precipitation." Many thanks.
* * *

---

## Author Response (AR1)

**Reply to reviewer #1 of "Sensitivity of warm clouds to large particles in measured marine aerosol size distributions – a theoretical study"**

Tom Dror[1], J. Michel Flores[1], Orit Altaratz[1], Guy Dagan[2], Zev Levin[3], Assaf Vardi[4], and Ilan Koren[1]

[1]Department of Earth and Planetary Sciences, Weizmann Institute of Science, Rehovot, Israel.

[2]Atmospheric, Oceanic and Planetary Physics, Department of Physics, University of Oxford, Oxford, UK.

[3]School of Earth Sciences, Department of Geophysics, Tel Aviv University, Ramat Aviv, Israel.

[4]Department of Plant and Environmental Sciences, Weizmann Institute of Science, Rehovot, Israel.

**We are grateful for the time and effort the reviewer invested in our work, and highly appreciate all of the constructive comments that helped us improve the paper. Below we address all the reviewer's comments point by point (our answers are marked in blue).**

In this paper, an axisymmetric cloud model with detailed bin–microphysics was initialized with six marine aerosol size distributions (MSD), measured in-situ in the Atlantic Ocean, Caribbean Sea and Pacific Ocean to study the effect of aerosol concentration and size distribution on warm clouds' properties. It shows that the cloud mass and precipitation change non-monotonically with the total concentration and size distribution of cloud condensation nuclei (CCN), especially when a tail of giant or ultragiant CCN (GCCN or UGCCN) is also included in the aerosol size distribution. The most interesting finding is the upper boundaries of the GCCN. This has not been reported in previous studies, to the best of my knowledge. The study is well within the scope of ACP and is generally well presented, except for a few places need to be clarified or corrected.

We thank the reviewer for the careful reading of our manuscript and this positive description of our work. We hope that the physical boundaries on GCCN will be helpful for the community.

**Specific comments:**

**1)** In the abstract, the statements and explanations are mainly based on the simulation results using the deepest thermodynamic profile, a more generalized statement or results including the shallower clouds should also be included, for a more complete picture.

**Authors reply:** We thank the reviewer for this important comment that helped us present our study in a more general way. We changed the abstract (and other parts in the paper, as described in answer no. 5) to describe the results of the different profiles.

The revised **abstract** reads: "Aerosol size distribution has major effects on warm cloud processes. Here, we use newly acquired marine aerosol size distributions (MSD), measured *in-situ* over the open ocean during the *Tara* Pacific expedition (2016—2018), to examine how the total aerosol concentration ($N_{tot}$) and the shape of the MSD change warm clouds' properties. For this, we used a toy-model with detailed bin-microphysics **initialized using three different atmospheric profiles, supporting the formation of shallow to intermediate and deeper warm clouds.** The changes in the MSDs affected the clouds' total

mass and surface precipitation. In general, the clouds showed higher sensitivity to changes in $N_{tot}$ than to changes in the MSD's shape, except for the case where the MSD contained giant and ultragiant cloud condensation nuclei (GCCN, UGCCN). For increased $N_{tot}$ **(for the deep and intermediate profiles),** most of the MSDs drove an expected non-monotonic trend of mass and precipitation **(the shallow clouds showed only the decreasing part of the curves with mass and precipitation monotonically decreasing).** The addition of GCCN and UGCCN drastically changed **the non-monotonic trend**, such that surface rain saturated and the mass monotonically increased with $N_{tot}$. GCCN and UGCCN changed the interplay between the microphysical processes by triggering an early initiation of collision-coalescence. The early fall-out of drizzle in those cases enhanced the evaporation below the cloud base. Testing the sensitivity of rain yield to GCCN and UGCCN revealed an enhancement of surface rain upon the addition of larger particles to the MSD, up to a certain particle size, when the addition of larger particles resulted in rain suppression. This finding suggests a physical lower bound can be defined for the size ranges of GCCN and UGCCN."

**2)** I suggest each of filled circles in Fig. 2 (c)(d) to be marked with a time. The current figures are a little bit confusing.

**Author reply:** We thank the reviewer for this comment that helped us make the figure clearer. In the revised Fig. 2 (see below), we marked the filled circles on the black and blue curves, to indicate the simulated time (in *min*).

We also added specific time references to the part of the revised *Results* where Fig. 2 is interpreted to make it easier to follow the trajectories in the figure, **(section 3, L157—L161):** "At a later stage in the cloud's lifetime, the trajectories turn diagonally up **(~56 *min* into the simulation)**, showing that the collection process has begun. Finally, the clouds stop growing by condensation, reaching their maximum mass, and begin to evaporate **(~71 *min* into the simulation;** trajectories turn to the left). In the *Atlantic—1* MSD case, the collection process kicks in earlier, within 10 minutes of the cloud's lifetime **(~51 *min* into the simulation)**, due to the presence of GCCN in the MSD which initially form bigger droplets."

[Figure]

*Figure 2.* **(a)** *Surface rain yield and* **(b)** *cloud's maximum mass as a function of* $N_{tot}$ *used in the simulation,* *integrated over 150 minutes of simulations. Each curve represents six simulations, done with a specific shape of the MSD normalized to different aerosol concentrations. The lower panels* **(c, d)** *show the time evolution of accumulated collected mass versus accumulated condensed mass.* **The simulated time is noted along the black and blue curves for the Atlantic—1 and Pacific—6 MSDs, respectively.** *The panels represent an aerosol concentration of 416 and 2629 cm*$^{-3}$ *(c and d, respectively).*

**3)** Line 183-184: "The *Atlantic-1* raindrops are considerably smaller than those produced by the other clouds (Fig. S6), and their evaporation is therefore greater": One fact might also be important is that while raindrops formed earlier in case *Atlantic-1*, the cloud is still in its developing stage, or the vertical velocity is still positive below cloud base, and the relative humidity is relatively low, so the raindrops spend more time and therefore evaporate more before reaching the surface. In other cases, rain was promoted by stronger downdrafts, and the relative humidity should also be higher. Therefore, it may provide more evidence to explain the differences in surface rain amount and evaporation mass between case *Atlantic-1* and other cases by analyze the below-cloud vertical velocity and relative humidity.

**Author reply:** We thank the reviewer for this important comment that allowed us to be more thorough in our explanation. We added to the SI a new figure (Fig. S7, attached below) that shows the time-height evolution of the horizontal mean values of cloud mass mixing ratio (Fig. S7a—b), droplet number concentration ($N_d$, Fig. S7c—d), vertical velocity ($w$, Fig. S7e—f), and relative humidity (*RH*, Fig. S7g—h)

for all the cloudy (and rainy) pixels of the *Atlantic—1* and *Pacific—6* MSDs for an aerosol concentration of 2629 *cm⁻³*. Note that the *RH* panels show only the sub—cloud layer to address the reviewer's comment. Focusing on the *w* and *RH* panels, they show clearly that the reduced surface rain of the *Atlantic—1* case is indeed due to a combination of the smaller raindrops, and their early fallout while the cloud is still in its developing stage. Therefore, the sub—cloud layer is dominated by updrafts and low *RH* values in comparison to the *Pacific—6* MSD (presented in this figure as a representative of all other MSDs).

We added an explanation to the *Results* **(section 3, L200—L205)**: "The *Atlantic—1* raindrops are considerably smaller than those produced by the other clouds (Fig. S6; see below), and their evaporation is, therefore, **more efficient. Moreover, the rain falls below the cloud base earlier, compared to the other MSDs cases, while the cloud is still in its developing stage, meaning that the cloud and the sub—cloud layers are dominated by updrafts, and the sub—cloud layer is consequently drier (Fig. S7). The combination of the small raindrops with their early fall out that lasts longer (due to the updrafts prevailing at this stage), results in greater rain evaporation below the cloud base for the *Atlantic—1* MSD.**"

We also added the following explanations to the *Summary* **(section 4, L265—L268):** "This results in **the** fast formation of large drops and the early fall-out of drizzle **while the cloud is still in its developing stage, such that updrafts prevail and the sub—cloud layer is drier. The combination of a sub—cloud layer that is dominated by updrafts and features lower *RH* values, further promotes longer fall time for the small raindrops and an efficient evaporation below the *Atlantic—1* cloud base.**"

**4)** Figure 4(e) and 4(f) show a little bit strange here and do not add more support to the main body of the text, may be removed?

**Author reply:** While we agree that these two images are exceptional in this paper, we believe they are a valuable addition. They emphasize the uniqueness of this work that relies on *in-situ* measurements as the initial conditions for the modeled aerosol's MSDs, while also showing the type and size of big aerosol (GCCN and UGCCN) that are present in the marine boundary layer. However, following the reviewer's comment and to better explain them we separated panels e—f from Fig. 4 and put them in a separate new Figure (Fig. 5, see below) in the *Summary* (section 4) of the revised manuscript, where we want to demonstrate which type of GCCN and UGCCN were measured over the open ocean.

[Figure]

*Figure S7. Time-height diagram of the horizontal mean of **(a, b)** cloud mass mixing ratio (g kg⁻¹), **(c, d)** droplet number concentration ($N_d$, cm⁻³), **(e, f)** vertical velocity (w, m s⁻¹), and **(g, h)** relative humidity (RH, %) below the cloud base, for the Altantic-1 (left column) and Pacific-6 (right column) MSDs normalized to an aerosol concentration of 2629 cm⁻³. Values are shown only for the cloudy (and rainy) pixels (mixing ratio > 10⁻³ g kg⁻¹). Note the different scales for the color bars in panels **(c)** and **(d)**.*

[Figure]

*Figure S6. Droplet size distribution below the cloud base at the time of maximum surface rain rate for the six different MSDs normalized to $N_{tot}$ = 2629 cm$^{-3}$. The total droplet number concentration ($N_d$, cm$^{-3}$) is noted in the legend for each MSD.*

**a) *Brochosome-like*      b) *Mineral dust**

[Figure]

*Figure 5. Scanning electron microscope images of Brochosome-like particles (a), and mineral dust (b) collected during the same period as the Atlantic-1 MSD measurement.*

**5)** In this paper, only results from the simulation with the most unstable thermodynamic profile are analyzed in detail. For shallower clouds, the rain yield and the max. cloud mass show monotonic change with CCN concentration and no significant changes with GCCN. So the conclusion should be generalized to reflect how the results change for other thermodynamic situations.

**Author reply:** We thank the reviewer for this comment. Based on this and to give a more general picture of our results, we changed the abstract to describe the other profile's results as well (see answer no. 1

above). In addition, we added more explanations regarding the intermediate and shallow profiles to other parts of the manuscript.

*Results* **(Section 3, L121—126):** "The general shape of the five curves is similar **for the deep and intermediate profiles**, and exhibits a non-monotonic trend (see Fig. 2a—b **and Fig. S2a—b, respectively**): an increase in total rain yield and the cloud's maximum mass as a function of aerosol loading, up to a maximum optimal aerosol concentration ($N_{op}$), followed by a decrease. All five curves have a similar $N_{op}$ of around $N_{tot} = 677 \, cm^{-3}$ **($N_{tot} = 416 \, cm^{-3}$ for the intermediate profile)** for both surface rain yield and maximum cloud mass. **For the shallow profile, the five MSD curves preset only the decreasing branch, with a minor decrease in rain yield and cloud mass with increasing aerosol loading.**"

*Results* **(Section 3, L135—139):** "For the cases of shallower cloudy-layers, where the clouds are more subjected to entrainment effects, the ascending branch of the curves is less pronounced (**intermediate profile,** Fig. S2a—b) or non-existent (**shallow profile,** Fig. S2c—d). We, therefore, focus on the deepest atmospheric profile, which better demonstrates the full effect of the competition and interactions between the microphysical processes in the clouds, **and refer to Text S2 in the SI for the intermediate and shallow profiles.**"

*Summary* **(Section 4, L250—254):** "We focused on the deepest profile, since it best captured the effect of competing microphysical cloud processes, and showed that surface rain yield and cloud's maximum mass are affected in a non-monotonic way by changes in $N_{tot}$, and the shape of the MSDs for most of the cases. **This was also the case for the intermediate profile results, while the shallow one only showed the decreasing branch of this non-monotonic trend, due to more dominant entrainment effects.**"

***Text S2. Additional Atmospheric Profiles*** in the SI was extended and now includes the following (L29—L36): "We examined the surface rain yield and the cloud's maximum mass as a function of $N_{tot}$. The results of the deepest cloud profile are shown in the main text (Fig. 2a—b), and the other two profiles are shown in Fig. S2. **The trends of the *Atlantic—1* surface rain yield and cloud's maximum mass curves for the intermediate profile are similar to the ones of the deeper profile. The only difference is that the rain yield values of the *Atlantic—1* are higher than the ones produced by the other MSDs for $N_{tot}$ > 677 $cm^{-3}$. All the curves show a lower $N_{op}$ compared to the deepest profile curves. Under the shallow thermodynamic profile, the *Atlantic—1* rain yield curve shows a similar trend to all other MSD cases, while producing the highest rain values. As for the trend in cloud mass, the *Atlantic—1* shows a monotonic increase (similar to the deep and intermediate profiles).**"

**Technical corrections:**

**1)** Line 143, change "accumulating" to "accumulated";

**Author reply:** Changed.

**2)** Line 206: Figure 4b,d should be "Figures 4b-d.

**Author reply:** Corrected.

**Reply to reviewer #3 of *"Sensitivity of warm clouds to large particles in measured marine aerosol size distributions – a theoretical study"**

Tom Dror[1], J. Michel Flores[1], Orit Altaratz[1], Guy Dagan[2], Zev Levin[3], Assaf Vardi[4], and Ilan Koren[1]

[1]Department of Earth and Planetary Sciences, Weizmann Institute of Science, Rehovot, Israel.

[2]Atmospheric, Oceanic and Planetary Physics, Department of Physics, University of Oxford, Oxford, UK.

[3]School of Earth Sciences, Department of Geophysics, Tel Aviv University, Ramat Aviv, Israel.

[4]Department of Plant and Environmental Sciences, Weizmann Institute of Science, Rehovot, Israel.

**We are grateful for the time and effort the reviewer invested in our work, and highly appreciate all of the constructive comments that helped us improve the paper. Below we address all the reviewer's comments point by point (our answers are marked in blue).**

The influence of aerosol size distribution and chemical composition on precipitation formation and intensity is still a challenging question to answer, due primarily to the sophisticated microphysical processes dealing with particles with a wide range of sizes, and also to the interplays between dynamics and microphysics. In this study, the authors choose to focus on addressing aerosol-precipitation response in a warm cloud, using a detailed bin microphysical framework for both aerosols and cloud droplets while a somewhat simplified dynamical framework (an axisymmetric model). In addition, they have also assumed a uniform chemical composition for the included aerosol population (sea salt) to limit the aerosol activation in a one-dimensional (size) parametric space. In order to address the targeted issue more realistically, they have also adopted measured aerosol size distributions collected from locations with different atmospheric backgrounds.

An interesting finding of this study is the significant difference in aerosol-precipitation responses between a case with the so-called *Atlantic-1* aerosol profile with ultra large CCNs and cases with other measured aerosol profiles without evident fraction of such giant CCNs. With a careful design of their modeling simulations, the authors have been able to define the criterion size of large aerosol particles that can create significant impacts on precipitation. Overall speaking, the paper has been relatively well organized, the research findings are well presented, and conclusions are drawn with solid science evidence.

We thank the reviewer for the careful reading of our manuscript and the constructive remarks.

**1)** A clear missing information in the manuscript is the cloud droplets concentrations, especially the vertical profiles of number concentration of cloud droplets and raindrops.

… In addition, Fig. 2(d) presents a rather interesting feature in high concentration simulations using all the distributions except *Atlantic-1* where collision-coalescence overwhelmed the condensation growth in a relatively early stage. However, without information of vertical distributions of cloud mass, the reader would have problem to understand (1) why the collision-coalescence increases with time but in

a rather slow pace comparing to the case of *Atlantic-1*, and (2) the depths of layer where cloud mass grew in various cases. Note that large droplets (i.e., raindrops) can still be moved upward by updraft and both condensation and collision-coalescence can proceed in either updraft or downdraft (as far as the parcel remains saturated). Therefore, knowledge of the vertical growth tracks of precipitating particles is critical to understand how the two major growing processes evolved.

**Author reply:** We thank the reviewer for these important comments. Here we address the reviewer's comments regarding the vertical profiles of number concentration of cloud droplets and raindrops, as well as the vertical distributions of cloud mass.

Following the reviewer's suggestion and to better explain our results, we added Text S7 and Figure S7 to the SI (see below) showing the time-height horizontal mean profiles of: cloud mass mixing ratio, droplet number concentration ($N_d$), vertical velocity ($w$), and relative humidity ($RH$) below the cloud base, for the *Atlantic—1* and *Pacific—6* MSDs, normalized to an aerosol concentration ($N_{tot}$) of 2629 $cm^{-3}$. We also added Text S8 and Figure S8 to the SI (see below as well) showing the time-height evolution of the horizontal mean profiles of the number concentration ($N_r$) and mass mixing ratio ($M_r$) of precipitating particles ($D_p > 80 \mu m$) for the *Atlantic-1* and *Pacific-6* MSDs, normalized to $N_{tot}$ = 2629 $cm^{-3}$.

We chose to show the *Pacific—6* MSD results as a representative example since it represents well all the other four MSDs cases. As can be seen in Fig. R1 below, that shows the time-height evolution of the horizontal mean profiles of cloud mass mixing ratio for all the MSDs for $N_{tot}$ = 2629 $cm^{-3}$, there are only minor differences among the four clouds.

We added an explanation regarding this issue to the *Results* (**section 3, L169—L181**): "Under more polluted conditions, the trajectory of the *Atlantic—1* MSD cloud (black curve in Fig. 2d) on this phase space is similar (in shape) to the one in the cleaner case (black curve in Fig. 2c), but this cloud accumulates more mass, due to the larger droplet surface area (Fig. S5b—c). **However, the *Atlantic—1*'s total droplet surface area is lower in comparison to the rest of the clouds (see Fig. S5c), and still, it condenses more mass, reaching $\sim 11 \times 10^7 g$ compared to $\sim 8 \times 10^7 g$ as the rest of the clouds. This can be explained by the nucleation of the GCCN and UGCCN that are present in the *Atlantic—1* MSD under polluted conditions, which on the one hand accumulate more mass (Fig. 2d) and drive a significantly higher number of raindrops at the growing stage of the cloud (Fig. S8), and on the other hand, results in a lower droplet number concentration ($N_d$) compared to the other clouds (Fig. S7). Therefore, the vertical distribution of mass of the *Atlantic—1* cloud is dominated by the precipitating particles, unlike the other clouds (Figs. S7 and S8). Note that while the total cloud mass of the *Atlantic—1* is larger than the one obtained by the other clouds, it is in the same order of magnitude. However, the mass of precipitating particles in the *Atlantic—1* cloud overwhelms the ones exhibited by the other MSDs. Contrastingly, the $N_d$ of the other clouds is much higher than the one of the *Atlantic—1* cloud, allowing for collision-coalescence to begin toward the end of the condensational growth stage, or after the evaporation process has begun (e.g., *Pacific—4*, the trajectories turn back to the left before acquiring a vertical component), and to increase in a slow pace. For the *Atlantic—1* cloud, the accumulation of liquid water by nucleation and condensation occurs in parallel to the collection process that starts much earlier in this case (Fig. 2d)."**

We added *Text S7. Time—height Diagrams of Cloud Mean properties* to the SI (L67—L76): "To understand the vertical distribution of some of the cloud's key properties, we show the time evolution of the cloud mass mixing ratio, droplet number concentration ($N_d$), vertical velocity ($w$), and relative humidity ($RH$) below the cloud base for the *Atlantic—1* and the *Pacific—6* MSDs normalized to $N_{tot}$ = 2629 $cm^{-3}$. We show the *Pacific—6* MSD case as a representative example to the other four MSD cases, since their results are very similar. It is clear that while the *Atlantic—1*'s mass is the same order of magnitude as the one of the *Pacific—6*, the total $N_d$ is considerably smaller for the *Atlantic—1* cloud, and that the droplets are confined to the lower part of the cloud. These are big droplets that nucleated on the GCCN and the UGCCN in the *Atlantic—1* MSD. These droplets sediment out almost immediately after their formation, thus are not carried to higher levels in the cloud. The *Atlantic—1* starts to precipitate earlier than the other clouds (as discussed in the main text), while the cloud is still in its developing stage, updrafts prevail and the sub—cloud layer features low $RH$ values."

We added *Text S8. Time—height Diagrams of Precipitating Particle's Growth* to the SI (L77—L84): "For clarifying the reasons behind the reduced surface rain amounts that are observed in the *Atlantic—1* MSD case, Fig. S8 shows the time-height evolution of the horizontal mean profiles of raindrops number concentration $N_r$ ($D_p$ > 80 $\mu m$), and mass mixing ratio $M_r$ ($D_p$ > 80 $\mu m$), for the *Atlantic—1* and the *Pacific—6* MSDs, normalized to $N_{tot}$ = 2629 $cm^{-3}$. The formation of raindrops is observed at a very early stage of the *Atlantic—1* cloud lifetime, compared to the timing of the rain formation in the *Pacific—6* case. In addition, it is clear that the high $N_r$ around the *Atlantic—1*'s cloud base contains very little mass, but the drops are big enough to fall out. However, since the drops are small, and the sub—cloud layer is still dominated by updrafts (Fig. S7e—f) the majority of them evaporate before reaching the surface (efficient evaporation and longer fall time)."

[Figure]

*Figure S7. Time-height diagram of the horizontal mean of **(a, b)** cloud mass mixing ratio (g kg$^{-1}$), **(c, d)** droplet number concentration (N$_d$, cm$^{-3}$), **(e, f)** vertical velocity (w, m s$^{-1}$), and **(g, h)** relative humidity (RH, %) below the cloud base, for the Atlantic—1 (left column) and Pacific—6 (right column) MSDs normalized to N$_{tot}$ = 2629 cm$^{-3}$. Values are shown only for the cloudy (and rainy) pixels (mixing ratio > 10$^{-3}$ g kg$^{-1}$). Note the different scales for the color bars in panels **(c)** and **(d)**.*

[Figure]

*Figure S8. Time—height diagram of the horizontal mean of raindrops ($D_p$ > 80 μm) **(a, b)** number concentration ($N_r$, $cm^{-3}$) and **(c, d)** mass mixing ratio ($M_r$, $g\ kg^{-1}$), for the Atlantic—1 (left column) and Pacific—6 (right column) MSDs normalized to $N_{tot}$ = 2629 $cm^{-3}$. Values are shown only for the cloudy (and rainy) pixels (mixing ratio > $10^{-3}$ $g\ kg^{-1}$). Note that the color bars have different limits for the Atlantic—1 and Pacific—6 clouds.*

[Figure]

*Figure R1. Time-height diagram of the horizontal mean profiles of cloud mass mixing ratio (g kg$^{-1}$) for all MSDs for an aerosol concentration of 2629 cm$^{-3}$. There are only minor changes among all MSD's besides the Atlantic—1. We, therefore, took Pacific—6 as a representative MSD.*

**2)** The authors have discussed the correlation between sub-cloud evaporation and rainfall at surface. With a knowledge of sub-cloud raindrop population including total number and size distribution this would be much easier to understand.

**Author reply:** Thank you for this comment. The revised Fig. S6 (see below) shows the droplet size distribution below the cloud base at the time of maximum surface rain rate for the six different MSDs normalized to $N_{tot}$ = 2629 cm$^{-3}$. The total number concentration of droplets was added for the different curves. The figure clearly shows that the *Atlantic—1* MSD has more small raindrops and less big raindrops than the other MSDs.

[Figure]

*Figure S6. Droplet size distribution below the cloud base at the time of maximum surface rain rate for the six different MSDs normalized to $N_{tot}$ = 2629 $cm^{-3}$. The total droplet number concentration ($N_d$, $cm^{-3}$) is noted in the legend for each MSD.*

**3)** It is understood that the authors wanted to focus on the aerosol and cloud microphysical connections. Nevertheless, the feedback of dynamics, even in a rather simplified dynamical framework still plays a role in determining the growth of precipitating particles. The authors mentioned very briefly about cold downdraft and also analyzed sub-cloud evaporation. Perhaps a more in-depth analysis would provide a better understanding of the role of dynamical feedback in, e.g., leading to the results presented in Fig. 3 and 4.

**Author reply:** We thank the reviewer for highlighting this important point. As described in answer no. 1 above, we are now showing the vertical velocity in the revised version of the SI in Fig. S7e—f. As the reviewer stated, the dynamics is strongly coupled to the microphysics, and hence it has a crucial effect on the growth of precipitating particles and now it is presented in the paper to create a full picture. The early fall-out of the smaller *Atlantic—1* rain drops, while the cloud is still in its developing stage, results in the *Atlantic—1* raindrops falling through updrafts that dominate the cloud and the sub-cloud layer at this stage (Fig. S7g—h). This causes a longer fall and together with the fact that the *Atlantic—1* sub—cloud layer is drier, promotes the efficient evaporation of the already smaller raindrops of the *Atlantic—1* cloud. This results in the reduced surface rain as shown in Figs. 3 and 4 of the main text.

To clarify this point in the main text, we added the following paragraph to the *Results* **(section 3, L200—L205)**: "The *Atlantic—1* raindrops are considerably smaller than those produced by the other clouds (Fig. S6; see below), and their evaporation is, therefore, **more efficient**. **Moreover, the rain falls below the cloud base earlier, compared to the other MSDs cases, while the cloud is still in its developing stage, meaning that the cloud and the sub—cloud layers are dominated by updrafts, and the sub—cloud**

**layer is consequently drier (see Fig. S7). The combination of the small raindrops with their early fall out that last longer (due to the updrafts prevailing at this stage), results in greater rain evaporation below the cloud base for the *Atlantic—1* MSD."**

We also added the following to the *Summary* **(section 4, L272—L275):** "This results in **the** fast formation of large drops and thus, an early fall-out of drizzle, **while the cloud is still in its developing stage, updrafts prevail and the sub—cloud layer is drier. The combination of a sub—cloud layer that is dominated by updrafts and features lower *RH* values, further promotes longer fall time for the small rain drops and an efficient evaporation below the *Atlantic—1* cloud base."**

**Some minor comments.**

**4)** Page 5, Figure 2(a) and (b): it would be helpful to provide the integration length of each simulation shown in these two figure panels in the figure caption.

**Author reply:** We added the integration length of the simulation (150 *min*) to the revised Fig. 2 caption.

**5)** Page 5, Ln 112: I understand the purpose of normalizing every distribution to match a given total concentration is for the convenience to identify the role of certain characteristics of size distributions such as shape in influencing the formation of precipitation. However, it is expected that the shift of the distributions to meet often much higher concentrations would increase the number of GCCN or even UGCCN. Could the authors provide such numbers even in the supplementary materials as a table or so? In addition, I don't remember this has been discussed in the manuscript, e.g., why the increase of GCCN still had no effect on the overall rain formation and growth for all cases including Atlantic-2 other than Atlantic-1.

**Author reply:** We thank the reviewer for this comment. As we normalize the MSDs to higher aerosol concentrations, we indeed shift the distributions such that they contain bigger particles (see Fig. S1 below). However, most of the MSDs (all except from the *Atlantic—1* and *Atlantic—2*) did not contain any GCCN or UGCCN even in the case of the highest $N_{tot}$. In the paper we define a physical threshold for GCCN ($D_p$~5 $\mu m$) and UGCCN ($D_p$~14 $\mu m$). These thresholds are marked on the revised panels of Fig. S1, for each $N_{tot}$. The *Atlantic—1* and the *Atlantic—2* are the only MSDs that contain GCCN, and the *Atlantic—1* is the only MSD that contains UGCCN. Even though the *Atlantic—2* contains some GCCN, they are present at low concentrations (*1.6x10$^{-6}$—7.5x10$^{-5}$ cm$^{-3}$*, for the lowest and highest $N_{tot}$, respectively). These concentrations of GCCN are low in comparison to e.g., the GCCN concentration of the *Atlantic—1* (*1.9x10$^{-5}$—5.7x10$^{-4}$ cm$^{-3}$*), and are not sufficient to cause any effect on the *Atlantic—2* rain formation and growth.

We added a clarification regarding this matter to *Methods* **(section 2.2., L109—111): "As we normalized the MSDs to higher values of $N_{tot}$, the MSDs shifted such that they contained larger particles (Fig. S1). However, only the *Atlantic—1* and *Atlantic—2* MSDs contained GCCN, and the *Atlantic—1* was the only MSD that contained UGCCN even in the case of the highest aerosol concentration."**

We also extended *Text S1. Normalized MSDs* to include the following paragraph **(L17—L21): "The *Atlantic—1* and the *Atlantic—2* are the only MSDs that contained GCCN, and the *Atlantic—1* is the only MSD that contained UGCCN. Note that even though the *Atlantic—2* contained some GCCN, they were present at low concentrations (*1.6x10^{-6}—7.5x10^{-5} cm^{-3}*, for the lowest and highest $N_{tot}$, respectively). These concentrations of GCCN are low in comparison to e.g., the GCCN concentration of the *Atlantic—1* (*1.9x10^{-5}—5.7x10^{-4} cm^{-3}*), and are not sufficient to cause any effect on the *Atlantic—2* rain formation and growth."**

**6)** Page 7, Ln 152: "bigger droplets resulted in a lower total droplets' surface area. . .", the sentence is somewhat ambiguous since such a result is not obvious, an explanation would be helpful here.

**Author reply:** We thank the reviewer for this comment.

We clarified this in the revised *Results* **(section 3, L161—L164):** "The bigger droplets **formed by the nucleation of GCCN** resulted in a lower droplets' surface area **for a given total water mass (compared to the one that would have formed from droplets nucleated on smaller CCNs)** for the *Atlantic-1* MSD case and also compared to the other MSDs (Fig. S5). **The lower total droplets' surface area** was then further reduced by the early initiation of the collection process."

[Figure]

*Figure S1: All of the MSDs used in the model. Each panel shows the MSDs normalized to the specific total aerosol concentration of the MSD noted in the lower left corner. The panels are organized from clean (**a**) to polluted (**f**) conditions. **Dotted and dash-dotted verticals line indicated the threshold for GCCN ($D_p$~5 µm) and UGCCN ($D_p$~14 µm), respectively.***

[revised manuscript text omitted]
^{-3}$ (Text S6; Fig. S6). Finally, we examine the time-height evolution of the horizontal mean profiles of key parameter of the clouds (Text S7–S8; Figs. S7–S8).

15 **Text S1. Normalized MSDs.** Six measured MSDs that represent a variety of different marine environments were normalized to the other five MSD concentrations (total of 36 MSDs). This allowed for a careful examination of the effect of both $N_{tot}$ and the MSD's shape. Figure S1 shows all MSDs. The *Atlantic–1* and the *Atlantic–2* are the only MSDs that contained GCCN, and the *Atlantic–1* is the only MSD that contained UGCCN. Note that even though the *Atlantic—2* contained some GCCN, they were present at low concentrations ($1.6 \times 10^{-6} - 7.5 \times 10^{-5} \, cm^{-3}$, for the lowest and highest $N_{tot}$, respectively). These

20 concentrations of GCCN are low in comparison to e.g., the GCCN concentration of the *Atlantic–1* MSD ($1.9 \times 10^{-5} - 5.7 \times 10^{-4} \, cm^{-3}$), and are not sufficient to cause any effect on the *Atlantic–2* rain formation and growth.

[Figure]

**Figure S1.** All of the MSDs used in the model. Each panel shows the MSDs normalized to the specific total aerosol concentration of the MSD noted in the lower left corner. The panels are organized from clean (a) to polluted (f) conditions. Dotted and dash-dotted verticals line indicate the threshold for GCCN ($D_p = 5 \, \mu m$) and UGCCN ($D_p = 14 \, \mu m$), respectively.

[revised manuscript text omitted]

**Text S6. Droplet Size Distribution Below the Cloud Base.** To understand the reduced surface rain caused by the enhanced evaporation below the cloud base of the *Atlantic–1* MSD shown in Fig. 3 of the main text, we calculated the mean droplet size distribution for the time of maximum rain for an area just below cloud base. Figure S6 shows the droplet size distribution for all of the MSDs for $N_{tot} = 2629\,cm^{-3}$. The biggest droplets in the *Atlantic–1* MSD case are about six orders of magnitude less in concentration compared to the other five MSDs, explaining the more efficient evaporation below cloud base.

[Figure]

**Figure S6.** Droplet size distribution below the cloud base at the time of maximum surface rain rate for the six different MSDs normalized to $N_{tot} = 2629\,cm^{-3}$. The total droplet number concentration ($N_d$, $cm^{-3}$) is noted in the legend for each MSD.

**Text S7. Time–height Diagrams of Cloud Mean properties.** To understand the vertical distribution of some of the cloud's key properties, we show the time evolution of the cloud mass mixing ratio, droplet number concentration ($N_d$), vertical velocity ($w$), and relative humidity ($RH$) below the cloud mass of the *Atlantic–1* and the *Pacific–6* MSDs normalized to $N_{tot} = 2629\,cm^{-3}$.
We show the *Pacific–6* MSD case as a representative example to the other four MSD cases, since their results are very similar. It is clear that while the *Atlantic—1*'s mass is the same order of magnitude as the one of the *Pacific–6*, the total $N_d$ is considerably smaller for the *Atlantic–1* cloud, and that the droplets are confined to the lower part of the cloud. These are big droplets that nucleated on the GCCN and the UGCCN in the *Atlantic–1* MSD. These droplets sediment out almost immediately after their formation, thus are not carried to higher levels in the cloud. The *Atlantic–1* starts to precipitate earlier than the other clouds (as discussed in the main text), while the cloud is still in its developing stage, updrafts prevail and the sub—cloud layer features low $RH$ values.

[Figure]

**Figure S7.** Time-height diagram of the horizontal mean of (a,b) cloud mass mixing ratio ($g\,kg^{-1}$), (c, d) droplet number concentration ($N_d$, $cm^{-3}$), (e, f) vertical velocity ($w$, $m\,s^{-1}$), and (g, h) relative humidity ($RH$, %) below the cloud base, for the *Atlantic–1* (left column) and *Pacific–6* (right column) MSDs normalized to $N_{tot} = 2629\,cm^{-3}$. Values are shown only for the cloudy (and rainy) pixels (mixing ratio $> 10^{-3} g\,kg^{-1}$). Note the different scales for the color bars in panels (c) and (d).

**Text S8. Time–height Diagrams of Precipitating Particle's Growth.** For clarifying the reasons behind the reduced surface rain amounts that are observed in the *Atlantic—1* MSD case, Fig. S8 shows the time-height evolution of the horizontal mean profiles of raindrops number concentration $N_r(D_p > 80\,\mu m)$, and mass mixing ratio $M_r(D_p > 80\,\mu m)$, for the *Atlantic–1* and the *Pacific–6* MSDs normalized to $N_{tot} = 2629\,cm^{-3}$. The formation of raindrops is observed at a very early stage of the *Atlantic–1* cloud lifetime, compared to the timing of the rain formation in the *Pacific–6* case. In addition, it is clear that the high $N_r$ around the *Atlantic—1*'s cloud base contains very little mass, but the drops are big enough to fall out. However, since the drops are small, and the sub—cloud layer is still dominated by updrafts (Fig. S7e–f) the majority of them evaporate before reaching the surface (efficient evaporation and longer fall time).

[Figure]

**Figure S8.** Time–height diagram of the horizontal mean of raindrops $(D_p > 80\,\mu m)$ (a, b) number concentration $(N_r, cm^{-3})$ and (c, d) mass mixing ratio $(M_r, g\,kg^{-1})$, for the *Atlantic–1* (left column) and *Pacific–6* (right column) MSDs normalized to $N_{tot} = 2629\,cm^{-3}$. Values are shown only for the cloudy (and rainy) pixels (mixing ratio $> 10^{-3}g\,kg^{-1}$). Note that the color bars have different limits for the *Atlantic–1* and *Pacific–6* clouds.